# Analysis of migration of pathogenic drug-resistant bacteria to soils and groundwater after fertilization with sewage sludge

Ewa Stańczyk-Mazanek👤*, Longina Stępniak

The Faculty of Infrastructure and Environment, Czestochowa University of Technology, Czestochowa, Poland

* e.stanczyk-mazanek@pcz.pl

## Abstract

The paper discusses the analysis of the effect of using sewage sludge for fertilization on the level of soil and groundwater contamination with drug-resistant bacteria. Other sanitary contaminants in these environments were also analysed. Composted sewage sludge was introduced into the sandy soil over a period of 6 months. The examinations were conducted under conditions of a lysimetric experiment with the possibility of collecting soil leachates (in natural conditions). The following doses of sewage sludge were used: 0, 10, 20, 30 and 40 t/ha calculated per experimental object containing 10 kg of sandy soil. The research were carried out within the time frame of one year. *Dactylis glomerata* grass was grown on the fertilized soils. In soils and leachates from soils (which may have polluted groundwater) collected from fertilized experimental objects, the sanitary condition and quantity of drug-resistant bacteria (mainly from the families *Enterobacteriaceae* and *Enterococcus*) were analysed one year after fertilization. Their drug resistance to selected antibiotics was also analysed based on current recommendations. The study showed that fertilization with sewage sludge (even after stabilization and hygienization) results in contamination of soil and infiltrating waters with many species of drug-resistant pathogenic bacteria. The lowest level of contamination of soil and water environment was found after the application of sewage sludge at a dose of 10 t/ha. The isolated drug-resistant strains of intestinal bacteria were less sensitive to older generations of antibiotics including cefazolin, ampicillin, and co-amoxiclav.

## 1. Introduction

The content of various drugs, including antibiotics, is often determined in the soil and aquatic environments. They are detected in surface water, groundwater and even in water purified and intended for consumption [1]. At the same time, new drug-resistant bacteria are often identified. They can also be found more and more often in food [2, 3]. The sources of these pollutants are mainly wastewater from pharmaceutical industrial plants and hospitals [4]. Hospital sewage is particularly dangerous [5, 6]. The sources of contamination also include domestic sewage, sewage sludge and waste from agriculture and animal husbandry [7–11].

In wastewater treatment plants, antibiotics are removed to an insufficient level [12, 13]. The consequence is the migration of antibiotics and their toxic metabolites to the aquatic and soil environment. Wastewater can be a source of contamination of soil, plants and water, including

**Data Availability Statement:** All relevant data are within the paper.

**Funding:** The scientific research was funded by the statute subvention of Czestochowa University of

Technology, Faculty of Infrastructure and Environment.

**Competing interests:** The authors have declared that no competing interests exist.

rivers and seas [14], with bacteria resistant to many drugs. Studies have documented the contamination of fish and shrimps with drug-resistant bacteria from the *Enterobacteriaceae* family [15].

Fertilization of soils with organic fertilizers and waste, including sewage sludge, may cause the spread of pathogenic microorganisms in the environment. There are reports on the presence of drug-resistant bacteria in sludge in the literature concerning sludge management [12, 16]. In both sludge and a conventional organic fertilizer provided by manure, numerous pathogenic forms have been found while they acquired or developed resistance mechanisms with respect to a number of antibiotics available in health care [17–19]. Particularly risky is the possible migration of these harmful forms with precipitation and water infiltrated to groundwater and further to drinking water intakes.

Sanitary cleanness of water, including groundwater, is one of the most important components of contemporary water management. However, many potentially hazardous microorganisms are often found in water [20].

Many studies also highlight the risk of environmental migration and the emergence of pathogenic forms of drug-resistant bacteria in water. One of the ways of contaminating groundwater can be organic waste fertilization, including the use of sewage sludge (which is a reservoir of various microbial species). It is therefore important to determine the risk of the natural use of this waste. The authors undertook a study to determine the effect of sewage sludge application at different doses on the sanitary status of soils and migration of pathogenic microorganisms, including *Enterobacteriaceae* and *Enterococcus* family bacteria, to fertilized soils and groundwater.

The presence of sanitary indicators, intestinal bacteria and enterococci was determined in the analyzed samples of sewage sludge used for fertilization, fertilized soils and waters infiltrating the soils. The authors analyzed the effect of using sewage sludge for fertilization (in doses 0, 10, 20, 30 and 40 t/ha) on the degree of soil and groundwater contamination with drug-resistant bacteria.

Studies have shown that there is a threat of migration of pathogenic bacteria, including drug-resistant ones, to enriched soils and groundwater after the natural use of sewage sludge. Of the four applied doses, only the lowest one (10 t/ha) did not cause sanitary contamination of the environment.

## 2. Material and methods

### 2.1. Research facilities

The examinations of the migration of drug-resistant microorganisms from sewage sludge to fertilized soil and then to leachate from the soil were conducted as part of a lysimetric experiment under condition of cultivation in a foil tunnel for one year.

Light soil with granulometric composition of loamy soil was used in the experiment. The pH value in the soil used in the lysimetric experiment was 6.4 and, according to the fertilization recommendations [21], it exhibited poorly acid reaction. Concentration of heavy metals in the soils determined according to the standards [21] and presented in Table 1 was below permissible levels recommended for fertilization with sewage sludge [22].

For fertilization purposes in the lysimetric experiment, the authors used the sewage sludge from two wastewater treatment plants situated in the south of Poland. Sewage sludge was formed after biological treatment of sewage using the activated sludge method. Then it was stabilized aerobically. After this process, sewage sludge was thickened mechanically and dewatered by means of a belt press. After dewatering of sewage sludge on belt presses, it was

**Table 1. Physical and chemical properties of soil and sewage sludge used in the lysimeter experiment.**

| Parameter | Unit | Soil | Sewage sludge |
|---|---|---|---|
| Organic matter | [% d. m.] | 0.8 | 47.0 |
| $pH_{H20}$ | - | 6.4 | 8.2 |
| Organic carbon | [g·kg$^{-1}$ d. m.] | 9.65 | 230 |
| N (total) | | 0.65 | 37.12 |
| P (available) | [mg·kg$^{-1}$d. m.] | 35.12 | 611.5 |
| K (available) | | 19.49 | 262.4 |
| Mg (available) | | 59.9 | 885.4 |
| Chromium (Cr) | | 1.6 | 19 |
| Zinc (Zn) | | 3.6 | 775 |
| Lead (Pb) | | 7.1 | 27 |
| Copper (Cu) | | 1.1 | 156 |
| Cadmium (Cd) | | 0.1 | 2.6 |
| Nickel (Ni) | | 0.92 | 120.1 |
| Mercury (Hg) | | 0.0018 | 0.52 |

d. m.—dry mass.

composted naturally in prisms (without lime additives) on plots on the site of the treatment plant for a period of 6 months.

The experiment was conducted in polyethylene lysimeters with capacity of 10 kg dedicated to sludge sampling. The lysimeters were filled with sandy soil and fertilised with different doses of sewage sludge. The objects were fertilized with the doses calculated per pot so that they corresponded to the amount of 10, 20, 30 and 40 tons of fertilizer per hectare. The control objects with non-fertilized soils were also used in the experimental procedure. The soil mixtures fertilized in this way were sown with *Dactylis glomerata* grass.

The humidity during the lysimetric experiment was maintained at the level of 60% of maximum water capacity by watering with well water. The soil (collected from lysimeters at the depth of 25 cm) and the leachates collected from the research objects were analysed one year after fertilization with sewage sludge. All the tests were carried out with 3 repetitions. The results are represented by mean levels from these repetitions.

Soil leachates can enter surface water and also cause their microbiological contamination.

## 2.2. Sanitary analysis of sewage sludge, fertilized soils and infiltrating waters

Sanitary examinations were conducted based on the recommendations for the analysed environments [23]. The sanitary analysis of water was carried out in accordance with the current standards used in Poland [24]. The content of mesophilic bacteria (potentially pathogenic—on a nutrient agar), *Salmonella* bacteria (on SS medium) [25] and *Enterobacteriaceae* and *Enterococcus* bacteria [26, 27] was determined in the sewage sludge used for fertilization, soil fertilized with sewage sludge and water leachates. The coliform index of *Escherichia coli* [28], *Clostridium perfringens* [29] and *Proteus vulgaris* [26, 27] was also determined.

## 2.3. Examinations of drug resistance of bacteria occurring in sewage sludge, soil and infiltrating waters

The collected research materials (sewage sludge, fertilized soil and water leachate from soils fertilized with sewage sludge) were also subjected to microbiological analyses, which were

aimed to determine the quantitative and qualitative composition of drug-resistant microorganisms on ENDO, BEA and nutrient agars (with three repetitions). Incubation of microorganisms was performed for 24 hours at 37˚C.

Identification of individual groups of isolated microorganisms used the respective selective agars mediums. Agar medium (MPA) was used to determine the total count of mesophilic (potentially pathogenic) microorganisms in the samples.

After bacteria colonies were grown, they were sieved for three times using the reduction method in order to obtain pure strains. Identification of isolated intestinal bacteria was based on biochemical Microgen GN-ID A + B multitests. Enterococci were identified by means of Microgen Strep ID biochemical multitests. As recommended by the manufacturer of the multitests, enterococci were incubated after inoculation for 24 hours at temperature of 37˚C. The results of determinations were analysed by means of Microgen MID 60 software.

Drug resistance of intestinal bacteria and enterococci isolated from the environments studies was examined using the diffusion-disc method. The Mueller-Hinton agar recommended in clinical diagnostics was used. In the case of intestinal bacteria, we performed the analysis of drug susceptibility to amikacin, co-amoxiclav, cefazolin, ceftazidime, cefuroxime, ciprofloxacin, ampicillin and gentamicin. In the case of determination of drug resistance in enterococci, tests were performed to examine susceptibility to ampicillin, ciprofloxacin, penicillin, erythromycin, streptomycin, vancomycin, chloramphenicol, tetracycline, linezolid and imipenem. The drugs used are most commonly used to combat these groups of microorganisms [26, 27].

After even spreading of the suspensions of the bacteria isolated on the Mueller-Hinton agar surface (on Petri dishes), the discs soaked with the respective antibiotics with recommended concentrations were applied and the incubation was performed at the temperature of 37˚C for one day. The results concerning susceptibility to individual antibacterial compounds were read from the interpretation tables for minimal inhibiting concentrations and the size of growth inhibition zones developed by the European Committee on Antimicrobial Susceptibility Testing [30].

## 3. Results

Physical, chemical and sanitary characterization of materials used in the study is presented in Tables 1 and 2.

Light sandy soil was used for the examinations Its reaction was 6.4 (Table 1) and, according to fertilizing recommendations [21], it exhibited poorly acid reaction. The content of chromium and mercury in the control soil was within the range of the permissible concentration in soil. Taking into account to the Institute of Soil Science and Plant Cultivation (IUNG) guidelines used to assess the degree of heavy metals contamination of soils with heavy metal, the contents of standardized metals such as zinc, lead, copper, cadmium and nickel in sandy soil used for fertilization in the lysimeter experiment could be determined as a natural quantity (0 degree of soil contamination) [31]. Concentration of heavy metals in the soils was below permissible levels recommended for fertilization with sewage sludge [22].

Table 2 presents the results of sanitary examinations used for lysimeter experiments of soil and sewage sludge.

The analysis of the results of the data presented in Table 2 showed that sandy soil did not contain sanitary contaminants. According to the recommendations [23], it could be classified as clean soil. The sewage sludge used in the experiment did not contain bacteria of the *Salmonella sp*. genus and could be used for fertilizing purposes.

**Table 2. Results of sanitary examinations of soil and sewage sludge used for the experiments.**

| The type of material to be tested | Determination | | | | | | |
|---|---|---|---|---|---|---|---|
| | Bacteria count | | | Total bacteria count [CFU/1ml] | | | |
| | *Escherichia coli* | *Clostridium perfringens* | *Proteus vulgaris* | Mesophilic | From the family *Enterobacteriaceae* | From the family *Enterococcus* | *Salmonella sp.* |
| Soil | n. d. | n. d. | $10^{-1}$ | $3.9 \cdot 10^3$ | n. d. | n. d. | n. d. |
| Sewage sludge | $10^{-6}$ | $10^{-5}$ | $10^{-5}$ | $20.1 \cdot 10^9$ | $1.2 \cdot 10^7$ | $6.5 \cdot 10^1$ | n. d. |

n. d.- not detected, CFU—colony forming unit.

Tables 3 and 4 present the results of sanitary tests of soil fertilized with sewage sludge and leachates from these soils. The results of quantitative analyses of mesophilic bacteria and *Enterobacteriaceae* and *Enterococcus* families obtained one year after fertilization are also presented.

Analysis of the results presented in Table 3 found that the fertilization with sewage sludge at all doses affected the sanitary condition of the fertilized sandy soil. Only non-fertilised control soil could be classified as sanitary clean. The amount of mesophilic bacteria determined in soils fertilized with 20, 30 and 40 t/ha suggests soil contamination. The total number of bacteria of above $2.5 \cdot 10^6$ according to the pattern for the assessment of soil sanitary condition indicated its pollution [23]. Similarly, the determination of other sanitary indices, including *Escherichia coli* indicates low contamination after application of doses of 10 and 20 t/ha and

**Table 3. Results of sanitary examinations of soil fertilized with different doses of sewage sludge (after completion of the experiment).**

| Dose of sewage sludge applied for soil fertilization [t/ha] | Determinations in soil fertilized with sewage sludge | | | | | |
|---|---|---|---|---|---|---|
| | Bacteria titer | | | Total bacteria count [CFU/1ml] | | |
| | *Escherichia coli* | *Clostridium perfringens* | *Proteus vulgaris* | Mesophilic | From the family *Enterobacteriaceae* | From the family *Enterococcus* |
| 0 | n. d. | n. d. | $10^{-1}$ | $2.4 \cdot 10^3$ | n. d. | n. d. |
| 10 | $10^{-1}$ | $10^{-2}$ | $10^{-3}$ | $3.5 \cdot 10^5$ | $5.2 \cdot 10^4$ | 2 |
| 20 | $10^{-1}$ | $10^{-2}$ | $10^{-4}$ | $4.5 \cdot 10^6$ | $3.7 \cdot 10^5$ | 6 |
| 30 | $10^{-3}$ | $10^{-3}$ | $10^{-4}$ | $6.2 \cdot 10^6$ | $4.3 \cdot 10^5$ | 9 |
| 40 | $10^{-4}$ | $10^{-3}$ | $10^{-5}$ | $5.53 \cdot 10^7$ | $7.5 \cdot 10^5$ | 12 |

n. d.—not detected, CFU—colony forming unit.

**Table 4. Results of sanitary examinations of water leachates from soil fertilized with different doses of sewage sludge (after completion of the experiment).**

| Dose of sewage sludge applied for soil fertilization [t/ha] | Determinations in water leachates from soil fertilized with sewage sludge | | | | | |
|---|---|---|---|---|---|---|
| | Bacteria titer | | | Total bacteria count [CFU/1ml] | | |
| | *Escherichia coli* | *Clostridium perfringens* | *Proteus vulgaris* | Mesophilic | From the family *Enterobacteriaceae* | From the family *Enterococcus* |
| 0 | n. d. | n. d. | n. d. | $2.2 \cdot 10^1$ | n. d. | n. d. |
| 10 | $10^{-2}$ | $10^{-1}$ | $10^{-1}$ | $1.6 \cdot 10^2$ | $1.5 \cdot 10^1$ | n. d. |
| 20 | $10^{-2}$ | $10^{-2}$ | $10^{-3}$ | $2.7 \cdot 10^3$ | $4.2 \cdot 10^2$ | 3 |
| 30 | $10^{-3}$ | $10^{-2}$ | $10^{-3}$ | $3.5 \cdot 10^3$ | $7.3 \cdot 10^3$ | 5 |
| 40 | $10^{-3}$ | $10^{-3}$ | $10^{-4}$ | $1.7 \cdot 10^4$ | $4.8 \cdot 10^3$ | 8 |

n. d.—not detected, CFU—colony forming unit.

strong contamination of fertilized soils after application of doses 30 and 40 t/ha. Similar observations were found for anaerobes *(Clostridium perfringens)*. There was also an increase in the number of spoilage bacteria in the fertilized soils, which was evidenced by the *Proteus vulgaris*. The use of sewage sludge also caused a significant increase in the number of intestinal bacteria from the *Enterobacteriaceae* family.

The results presented in Table 4 prove that microorganisms from fertilizing materials, including sewage sludge, are likely to migrate to soils and further to groundwater. The increase of mesophilic (potentially pathogenic) and intestinal *Enterobacteriaceae* bacteria was determined in the leachates from sandy soil fertilized with sewage sludge. The lowest contamination was found after the application of the dose of 10 t/ha.

Table 5 presents the results of tests of resistance to selected antibiotics of bacterial species isolated from sewage sludge. Tables 6–8 show the results of drug resistance of bacteria isolated from soils and water leachates (from soil mixtures) fertilized with sewage sludge.

The data in Tables 2 and 5 show that *Enterobacteriaceae* are the most abundant intestinal bacteria in sewage sludge. A smaller group of bacteria are *Enterococcus* family. Few species were isolated from this group of organisms: *E. faecalis*, *E. faecium* and *E. gallinarum* (Table 8). All three species were found in sewage sludge and fertilized soil. The species migrating to and

**Table 5. Results of antibiograms for individual bacteria from the *Enterobacteriaceae* family and saprophytic bacteria isolated from sewage sludge.**

| Bacteria species isolated from sewage sludge | Type of antibiotic used | | | | | | | |
|---|---|---|---|---|---|---|---|---|
| | Co-amoxiclav (AMC30) | Amikacin (AK30) | Gentamicin (CN10) | Cefazolin (CZ30) | Ciprofloxacin (CIP5) | Ceftazidime (CAZ30) | Ampicillin (AM10) | Cefuroxime (CXM30) |
| *Citrobacter freundii* | S | S | S | S | S | S | MS | S |
| *Morganella morganii* | R | S | S | R | S | S | R | MS |
| *Klebsiella pneumoniae* | S | S | S | S | S | S | MS | S |
| *Klebsiella oxytoca* | S | R | S | S | S | S | R | S |
| *Yersinia aldovae* | S | S | S | S | S | S | R | S |
| *Yersinia enterocolitica* | R | S | R | S | S | R | R | S |
| *Serratia marcescens* | S | S | S | R | S | S | S | S |
| *Serratia rubidaea* | S | S | S | S | S | S | MS | S |
| *Burkholderia pseudomallei* | R | S | S | S | S | S | R | S |
| *Pseudomonas fluorescens** | R | S | S | S | S | S | S | S |
| *Pseudomonas stutzeri** | S | S | S | S | S | S | R | S |
| *Alcaligenes faecalis** | S | S | R | S | S | S | S | R |
| *Photorhabdus luminescens* | S | S | S | S | S | S | R | S |
| *Proteus vulgaris* | S | S | S | R | S | MS | R | R |
| *Providencia rettgeri* | S | S | S | S | S | S | R | S |
| *Providencia stuartii* | R | S | S | R | S | MS | R | R |
| *Enterobacter kobei* | S | MS | S | S | S | S | R | S |
| *Eschericha coli* | S | S | S | S | S | S | S | S |
| *Eschericha coli* | R | S | S | S | S | S | R | S |
| *E. coli—inactive L+* | R | S | S | S | S | S | R | S |
| *E. coli—inactive L-* | S | S | S | S | S | S | R | S |
| *Enteric Group* | R | S | S | S | S | S | R | S |

Symbols used for susceptibility of bacteria to antibiotics: S—susceptible, MS—medium susceptible, R—resistant.

* species that are typically saprophytic in the environment.

**Table 6. Results of antibiograms for individual bacteria from the *Enterobacteriaceae* family and saprophytic bacteria isolated from soil after a year from fertilization with sewage sludge.**

| Bacteria species isolated from soil | Dose of sewage sludge applied for soil fertilization [t/ha] ** | Type of antibiotic used | | | | | | | |
|---|---|---|---|---|---|---|---|---|---|
| | | Co-amoxiclav (AMC30) | Amikacin (AK30) | Gentamicin (CN10) | Cefazolin (CZ30) | Ciprofloxacin (CIP5) | Ceftazidime (CAZ30) | Ampicillin (AM10) | Cefuroxime (CXM30) |
| *Citrobacter freundii* | 20,30,40 | S | S | S | S | S | S | R | S |
| *Morganella morganii* | 10,20,30,40 | R | S | S | R | S | S | R | MS |
| *Klebsiella pneumoniae* | 10,20,30,40 | S | S | S | S | S | S | MS | S |
| *Klebsiella oxytoca* | 30,40 | S | MS | S | S | S | S | R | S |
| *Yersinia enterocolitica* | 10,20,30,40 | R | S | R | S | S | R | R | S |
| *Serratia marcescens* | 10,20,30,40 | S | S | S | R | S | S | MS | S |
| *Serratia rubidaea* | 20,30,40 | S | S | S | MS | S | S | MS | S |
| *Burkholderia pseudomallei* | 20,30,40 | R | S | S | S | S | S | R | S |
| *Pseudomonas fluorescens** | 10,20,30,40 | MS | S | S | S | S | S | S | S |
| *Alcaligenes faecalis** | 20,30,40 | S | S | R | S | S | S | R | R |
| *Eschericha coli* | 10,20,30,40 | S | S | S | S | S | S | S | S |
| *Eschericha coli* | 30,40 | MS | S | S | S | S | S | R | S |
| *E. coli—inactive L+* | 10,20,30,40 | R | S | S | S | S | S | R | S |
| *E. coli—inactive L-* | 10,20,30,40 | S | S | S | S | S | S | R | S |
| *Enteric Group* | 20,30,40 | MS | S | S | S | S | S | R | S |

Symbols used for susceptibility of bacteria to antibiotics: S—susceptible, MS—medium susceptible, R—resistant.

* species that are typically saprophytic in the environment.

** the type of the dose used for fertilization in which the microorganism was found to be present.

determined in soil was *E. faecalis*. An increase in the number of these microorganisms in the soil with an increase in the fertilization dose was observed. Sanitary parameters determined in soil leachates indicated significant exceeding of parameters in relation to waters used for drinking [32] and bathing purposes. According to the standards, water used e.g. for bathing purposes should not contain more than 100 CFU per 100 ml (recommended value) *of Escherichia coli* or fecal coliform bacteria (thermotolerant coliforms) and fecal streptococci (enterococci).

After the application of doses of 20, 30 and 40 t/ha of sewage sludge to the soil, the level of water infiltrating the fertilized soil did not meet the above requirements. A worrying phenomenon was the observed resistance of *E. faecalis* bacteria to erythromycin (Table 8).

The examinations showed that sewage sludge is a significant source of drug-resistant *Enterobacteriaceae* bacteria in organic soils fertilized with this waste (Tables 5 and 6). It was also shown that these microorganisms may migrate together with soil leachate to groundwater, posing a real threat to the environment (Table 7). The data in Table 7 also show that fertilization with the lowest dose (10 t / ha) of sewage sludges did not cause significant contamination

**Table 7. Results of antibiograms for individual bacteria from the *Enterobacteriaceae* family and saprophytic bacteria isolated in water leachates from soil fertilized with sewage sludge.**

| Bacteria species isolated from water | Dose of sewage sludge applied for soil fertilization [t/ha] ** | Type of antibiotic used | | | | | | | |
|---|---|---|---|---|---|---|---|---|---|
| | | Co-amoxiclav (AMC30) | Amikacin (AK30) | Gentamicin (CN10) | Cefazolin (CZ30) | Ciprofloxacin (CIP5) | Ceftazidime (CAZ30) | Ampicillin (AM10) | Cefuroxime (CXM30) |
| *Morganella morganii* | 20,30,40 | R | R | S | R | S | S | R | MS |
| *Klebsiella pneumoniae* | 20,30,40 | S | S | S | S | S | S | MS | S |
| *Klebsiella oxytoca* | 20,30,40 | S | S | S | S | S | S | R | S |
| *Yersinia aldovae* | 30,40 | S | S | S | S | S | S | R | S |
| *Yersinia enterocolitica* | 20,30,40 | R | S | R | S | S | R | R | S |
| *Serratia marcescens* | 20,30,40 | S | S | S | R | S | S | S | S |
| *Serratia rubidaea* | 30.40 | S | S | S | S | S | S | MS | S |
| *Pseudomonas fluorescens** | 10,20,30,40 | R | S | S | S | S | S | S | S |
| *Alcaligenes faecalis** | 20,30,40 | S | S | R | S | S | S | R | MS |
| *Providencia rettgeri* | 30,40 | S | S | S | S | S | S | MS | S |
| *Eschericha coli* | 10,20,30,40 | S | S | S | S | S | S | S | S |
| *Eschericha coli* | 20,30,40 | MS | S | S | S | S | S | R | S |
| *E. coli—inactive L+* | 20,30,40 | R | S | S | S | S | S | R | S |
| *E. coli—inactive L-* | 20,30,40 | S | S | S | S | S | S | MS | S |
| *Enteric Group* | 10,20,30,40 | R | S | S | S | S | S | R | S |

Symbols used for susceptibility of bacteria to antibiotics: S—susceptible, MS—medium susceptible, R—resistant.

* species that are typically saprophytic in the environment.

** the type of the dose used for fertilization in which the microorganism was found to be present.

of sandy soil and water leachate with drug-resistant pathogenic bacteria from *Enterobacteriaceae* family. The use of higher doses of sewage sludges increased of soil and water pollution with these microorganisms.

## 4. Discussion

Analysis of the data presented in Tables 5–8 indicates a significant risk of various diseases in humans and animals in contact with sewage sludge used in the experiment, fertilized soils and soil leachates. The sanitary condition of the sewage sludge depends on the type of technological processes used to treat wastewater. The most frequent processes used to limit the amount of pathogens include methane fermentation of both wastewater and sewage sludge [33, 34]. Further hygienization of this waste can be achieved by e.g. liming or composting [35].

In the case of the bacteria from *Enterobacteriaceae* and *Enterococcus* families determined in the examinations of sanitary indices, their presence was demonstrated in the analysed samples of sewage sludge, fertilized soil and water leachates. Intestinal bacteria count (*Enterobacteriaceae*) was at a worrying high level. Presence of enterococci was also found (Table 8). However,

**Table 8. Results of antibiograms for bacteria from the *Enterococcus* genus isolated from the sewage sludge, soil fertilized with sewage sludge and water leachates.**

| Species of isolated bacteria | | Chloramphenicol (C30) | Ciprofloxacin (CIP5) | Ampicillin (AM10) | Erythromycin E15 | Penicillin (P10) | Streptomycin (S300) | Linezolid (LNZ30) | Tetracycline (TE30) | Vancomycin (VA30) | Imipenem (IMP10) |
|---|---|---|---|---|---|---|---|---|---|---|---|
| Sewage sludge | *E. faecalis* | S | S | S | R | MS | R | R | S | S | S |
| | *E. faecium* | S | S | S | MS | MS | S | S | S | S | S |
| | *E. gallinarum* | S | S | S | MS | MS | S | S | S | MS | S |
| Soil (doses 30 and 40 t/ha) ** | *E. faecalis* | S | S | S | R | MS | R | MS | S | S | S |
| | *E. faecium* | S | S | S | MS | MS | S | S | S | S | S |
| | *E. gallinarum* | S | S | S | MS | MS | S | S | S | S | S |
| Water leachates (doses 40 t/ ha) ** | *E. faecalis* | S | S | S | R | MS | MS | MS | S | S | S |

Symbols used for susceptibility of bacteria to antibiotics: S—susceptible, MS—medium susceptible, R—resistant.

** the type of the dose used for fertilization in which the microorganism was found to be present.

only very small contents were observed for these bacteria, especially in the soil fertilized by sewage sludge.

Furthermore, no excessive migration to groundwater was found with increasing the dose of the sludge (Tables 3 and 4). Therefore, they do not represent a major epidemiological problem in the examined environments according to the recommended standards. Only *E. faecalis* migrated to soil leachates fertilized with sewage sludge from three bacterial species of the *Enterococcus* family determined in sewage sludge. Despite a low count of bacteria from this species, their resistance to erythromycin and mean resistance to penicillin, streptomycin and linezoid were determined. Similar results were obtained by Da Silva et al. [36]. In the raw wastewater, these researchers found *Entercoccus hirae*, *Entercoccus faecium* and *Entercoccus faecalis*. A decrease in the count of *Entercoccus hirae* bacteria and an increase in the count of *Entercoccus faecium* and *Entercoccus faecalis* were found in the treated municipal wastewater. Both species were characterized by 40% resistance to erythromycin. Drug-resistant strains of enterococci were not eliminated through wastewater treatment. Da Costa et al. also isolated *Enterococcus* bacteria in sewage and sewage sludge which showed 24.8% resistance to erythromycin. In 49.4%, they also showed multi-drug resistance [37]. These are bacteria that represent a serious epidemiological threat [38, 39].

In this study, the doses of sewage sludge of 10, 20, 30 and 40 t/ha did not cause contamination by these microorganisms. However, there is a risk that microbiological contamination with drug-resistant forms may already be significant after the application of higher doses.

The group of intestinal bacteria from the *Enterobacteriaceae* family dominated in the sewage sludge used for fertilization (Table 2). Among them, many isolated species (Table 5) have shown resistance to the antibiotics used. These are the drugs that are most often used to combat these forms of microorganisms. Ampicilin turned out to be the least effective drug for the determined bacteria. Among the isolated species, 68.2% showed resistance to this drug, while 13.6% showed average resistance. Only 18.2% of the intestinal bacteria tested in the sewage sludge were sensitive to this antibiotic. Similarly high ampicilin resistance in *Enterobacteriaceae* was observed in a study by Mahmud et al. [40].

Analysis of the data of the content of pathogenic drug-resistant bacteria of *Enterobacteriaceae* family and saprophytes in sewage sludge (Table 5) and comparison with isolated microorganisms from the soil fertilized with them (Table 6) and in infiltrating ground water (Table 7) revealed that the microbial forms tested were actively migrating in the environment. No pathogenic forms were isolated from the control soil. The application of sewage sludge doses above 10t/ha caused the appearance of drug-resistant *Escherichia coli* (a conditional pathogen) and other intestinal bacteria in the fertilized soil and ground water. It is noteworthy that pathogenic drug-resistant bacteria of the *Klebsiella* genus were isolated from both sewage sludge and the soils and groundwater fertilized with it. Some *Klebsiella* species are a natural part of the human intestinal flora and of the oral and nasal cavities. They represent opportunistic pathogens. The *Klebsiella pneumoniae* bacteria (sometimes called a superbug) can cause several serious illnesses (acute pneumonia often leading to death, urinary tract infection, and even sepsis) in weakened organisms. Many researchers describe it as highly resistant to most antibiotics (some scientists even believe that they are resistant to all antibiotics, as is the case with the New Delhi strain). Another dangerous bacterium isolated from soil and groundwater (after application of sewage sludge doses higher than 20 t/ha) was *Klebsiella oxytoca*. This bacterium may be responsible for endocolitis and sepsis [20].

Particularly dangerous are *Klebsiella oxytoca* species, considered to be alert bacteria. They can cause haematosepsis and various infections. Its resistance to the amikacin and ampicillin used in the experiments may be dangerous. These organisms may migrate to soils fertilized with sewage sludge and further to groundwater, which was confirmed in the study. This

phenomenon was particularly observed after the application of doses of sewage sludge of 30 and 40 t/ha. A similar phenomenon was observed in the case of the equally dangerous *Klebsiella pneumoniae* bacteria. Both species very quickly and easily become resistant to most antibiotics. Other experiments performed by the authors [18] also indicated the likelihood of presence and migration of these bacteria from the sewage sludge to soil, especially if higher doses of sewage sludge are applied. An increase in the degree of resistance of isolated bacteria was also observed at that time.

In our research, 36.6% of the isolated bacteria from the *Enterobacteriaceae* family and saprophytic bacteria isolated from sewage sludge, soil and water were resistant to co-amoxiclav. Ciprofloxacin (100% sensitivity) proved to be the most effective antibiotic against the determined microorganisms.

Among the isolated bacteria from the *Enterobacteriaceae* family, the species *Escherichia coli* occurred in large counts. The studies found different groups of these microorganisms that differed in reaction to the antibiotics used. Some were entirely sensitive to all drugs. Others, however, showed resistance mainly to co-amoxiclav and ampicillin. Differences in the reaction were also observed in relation to the presence of unstable forms of L+ and L- among isolated *E. coli* bacteria. Some authors [41] have demonstrated that L-forms (with cells of this shape), also called wild strains, are resistant to penicillin. The studies also showed resistance to ampicillin.

Some *E. coli* bacteria can contaminate plant biomass through soil and water. Multi-drug-resistant strains are particularly dangerous [42]. Other authors found the presence of other drug-resistant *Acinetobacter* bacteria in the biomass of lettuce and fruit [43].

Other dangerous bacteria isolated from the sewage sludge that migrated to soil and groundwater were *Serratia marcescens* and *Serratia rubidaea*. *S. marcescens* bacteria were initially considered non-hazardous saprophytic microorganisms, occurring mainly in aquatic environments. The first documented cases of human infection were found in the area of Great Britain at the beginning of the 20th century. Since then, further reports of urinary tract and endocardial infections, meningitis and sepsis have also been documented. Some researchers demonstrated a substantial intensification of the problem of hospital-acquired conditions caused by *S. marcescnes* in neonatal and paediatric wards. Depending on the centre studied, *S. marcescens* bacteria account for between 5 and 16% of nosocomial infections among newborns and infants [44]. The species *S. marcescens* are microorganisms resistant to numerous groups of antibiotics. Resistance to cefazolin was found in the present study. The migration of these bacteria to soils fertilized with sludge and groundwater seems to be worrying.

*Yersinia* bacteria were detected in sewage sludge, fertilized soils and infiltrating water. Isolated species of *Y. enterocolitica* and *Y. aldovae* can cause many diseases. *Y. enterocolitica* bacteria cause yersiniosis, systemic infections and haematosepsis, which are dangerous for human health. Isolated in the study, these bacteria turned out to be resistant to many antibiotics including: co-amoxiclav, gentamicin, ceftazidime and ampicillin.

## 5. Conclusions

1. The results of the study showed that the fertilization with sewage sludge at all doses affected the sanitary condition of the fertilized sandy soil. Only non-fertilised control soil could be classified as sanitary clean. The lowest level of contamination was found after application of the dose of sewage sludge of 10 t/ha.

2. The research indicates a significant problem of the risk of occurrence of various diseases in humans and animals in contact with sewage sludge and with soil and soil leachates fertilized

with this sludge. Many drug-resistant bacteria were isolated from the material studied, including *Klebsiella oxytoca*, which is considered to be an "alert bacterium".

3. The results show that pathogenic microorganisms, including those drug-resistant, can migrate from sewage sludge to soil and further to groundwater.

4. Most often isolated drug-resistant strains of intestinal bacteria were less sensitive to older types of antibiotics, including cefazolin, ampicillin and co-amoxiclav.

5. Resistance of several isolated strains of intestinal bacteria to newer antibiotics, e.g. ceftazidime is worrying. However, the determined bacteria were not resistant to ciprofloxacin.

## Author Contributions

**Conceptualization:** Ewa Stańczyk-Mazanek.

**Data curation:** Ewa Stańczyk-Mazanek, Longina Stępniak.

**Formal analysis:** Ewa Stańczyk-Mazanek, Longina Stępniak.

**Funding acquisition:** Ewa Stańczyk-Mazanek, Longina Stępniak.

**Investigation:** Ewa Stańczyk-Mazanek.

**Methodology:** Ewa Stańczyk-Mazanek, Longina Stępniak.

**Project administration:** Ewa Stańczyk-Mazanek.

**Resources:** Ewa Stańczyk-Mazanek.

**Software:** Ewa Stańczyk-Mazanek.

**Supervision:** Ewa Stańczyk-Mazanek.

**Validation:** Ewa Stańczyk-Mazanek, Longina Stępniak.

**Visualization:** Ewa Stańczyk-Mazanek.

**Writing – original draft:** Ewa Stańczyk-Mazanek, Longina Stępniak.

**Writing – review & editing:** Ewa Stańczyk-Mazanek.

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
