## [Decision Letter · Decision Letter 0]

17 Sep 2021

PONE-D-21-26629Analysis of migration of pathogenic drug-resistant bacteria to soils and groundwater after fertilization with sewage sludgePLOS ONE

Dear Dr. Stańczyk-Mazanek,

Thank you for submitting your manuscript to PLOS ONE. After careful consideration, we feel that it has merit but does not fully meet PLOS ONE’s publication criteria as it currently stands. Therefore, we invite you to submit a revised version of the manuscript that addresses the points raised during the review process.Please consider the comments of both reviewers carefully and address them as much as possible as they are required for acceptance.Please pay attention to the reviewers' comments on the abstract and revise the abstract accordingly so that it contains all crucial information, especially key results obtained and the conclusions drawn from the results. Please submit your revised manuscript by Nov 01 2021 11:59PM. If you will need more time than this to complete your revisions, please reply to this message or contact the journal office at plosone@plos.org. Please include the following items when submitting your revised manuscript:A rebuttal letter that responds to each point raised by the academic editor and reviewer(s). You should upload this letter as a separate file labeled 'Response to Reviewers'.A marked-up copy of your manuscript that highlights changes made to the original version. You should upload this as a separate file labeled 'Revised Manuscript with Track Changes'.An unmarked version of your revised paper without tracked changes. You should upload this as a separate file labeled 'Manuscript'.

We look forward to receiving your revised manuscript.

Kind regards,

Reginald B. Kogbara, Ph.D.

Academic Editor

PLOS ONE

Journal Requirements:

2. In your Methods section, please provide additional details regarding the experimental materials (soils, sludge) used in your study and ensure you have described the source. For more information regarding PLOS' policy on materials sharing and reporting, see https://journals.plos.org/plosone/s/materials-and-software-sharing#loc-sharing-materials.

"The scientific research was funded by the statute subvention of Czestochowa University of Technology, Faculty of Infrastructure and Environment"

"The scientific research was funded by the statute subvention of Czestochowa University of Technology, Faculty of Infrastructure and Environment"

"The scientific research was funded by the statute subvention of Czestochowa University of Technology, Faculty of Infrastructure and Environment"

7. PLOS requires an ORCID iD for the corresponding author in Editorial Manager on papers submitted after December 6th, 2016. Please ensure that you have an ORCID iD and that it is validated in Editorial Manager. To do this, go to ‘Update my Information’ (in the upper left-hand corner of the main menu), and click on the Fetch/Validate link next to the ORCID field. This will take you to the ORCID site and allow you to create a new iD or authenticate a pre-existing iD in Editorial Manager. Please see the following video for instructions on linking an ORCID iD to your Editorial Manager account: https://www.youtube.com/watch?v=_xcclfuv

Reviewers' comments:

Reviewer's Responses to Questions

**Comments to the Author**

1. Is the manuscript technically sound, and do the data support the conclusions?

Reviewer #1: Partly

Reviewer #2: Yes

2. Has the statistical analysis been performed appropriately and rigorously? 

Reviewer #1: N/A

Reviewer #2: N/A

3. Have the authors made all data underlying the findings in their manuscript fully available?

Reviewer #1: Yes

Reviewer #2: Yes

4. Is the manuscript presented in an intelligible fashion and written in standard English?

Reviewer #1: Yes

Reviewer #2: Yes

5. Review Comments to the Author

Reviewer #1: The article entitled "Analysis of migration of pathogenic drug-resistant bacteria to soils and ground water after fertilization with sewage sludge" aimed to study the effect of using sewage sludge as fertilizer for contamination with drug-resistant bacteria in the environment.

In my opinion, the article needs major revision for publication in PLOS one as the structure and quality of the manuscript is not up to quality. The experiment used a novel approach but fail to design and represent the study in a scientifically coherent manner.

The abstract is not organized in a coherent manner. There is no proper introduction and concluding remark. Author should provide a summary of what is known, the rationale for the study, the method used, the outcome obtained, and conclusions drawn.

The introduction should focus on more how sewage sludge led to migration of pathogenic drug-resistant bacteria, the consequence and why the experiment need to be done

The materials and methods should be written in a way of how experiment is done and condensed by removing description not related to materials and methods.

In table 3 and 4, I am not sure why increased dose of sewage sludge applied for soil fertilization would lead to decreased CFU of bacterial strain such as E. coli.

For Table 5, 6 7 and 8, it does not make any sense to evaluate the antimicrobial resistant pattern of a single strain as it will never represent the while scenario. Author should test more random colony from the same sample to make a logical conclusion. Author should find a way to compare results of table 5, 6 and 7 together.

The results should be quantitatively compared in conclusion with other published data.

Reviewer #2: Analysis of migration of pathogenic drug-resistant bacteria to soils and groundwater after fertilization with sewage sludge

The study entitled “Analysis of migration of pathogenic drug-resistant bacteria to soils and groundwater after fertilization with sewage sludge” deals with one of the most alarming threat connected with the environmental safety and human and animal health. The strong side of this article is the duration devoted to the experiment and the genus identification of the obtained strains with antibiotic-resistance analysis in the further part of research. All the needed data is acquired in the manuscript. The manuscript is well organized and it is carrying the proper scientific soundness. However, in some places, there is no precise description of results, errors in discussion, and vocabulary mistakes throughout the manuscript. Here follows the detailed questions and concerns which could help with correction of the manuscript.

Abstract line 13-23:

The Abstract contained well described aim of the study and used methods, but clearly it is lacking the brief characterization of most important results.

Materials and Methods line 99-100

Authors based sanitary analysis on Polish laboratory exercises book (references number 23), in my opinion this could be not sufficient. I encouraged authors to make references on what detailed regulation they based their methods.

Materials and Methods line 135

I think, it is supposed to be “discs soaked” not “dishes soaked”

Results Table 1. line 168-171

In my opinion the Table 1 structure it can be difficult to read which Unit stands for what Parameter. I encouraged to add some additional horizontal lines.

Results line 223-225

In the sentence “According to the standards, water used e.g. for bathing purposes should not contain more than 100 CFU per 100 ml.”, in my opinion this should be clarified to which group of bacteria it refers.

Results line 231-233

The last sentence of the paragraph is unclear and it could be clarified with references to specific Table/Tables.

Results Table 6. and Table 7. line 240 and 252

In those tables I found the column named “Dose of sewage sludge applied for soil fertilization [t/ha]” with it values very unclear. Is this values represent sewage sludge concentrations in which resistant bacteria occurred ? In my opinion additional description should appear.

Discussion line 317-320

Please verify if defined percentage values are correct.

Discussion line 322

Please specify in which samples or stage of the experiment 36.6% of isolated bacteria were resistant to co-amoxiclav.

Manuscript

Please verify the units correctness with SI Units convention.

6. PLOS authors have the option to publish the peer review history of their article (what does this mean?). If published, this will include your full peer review and any attached files.

Reviewer #1: No

Reviewer #2: No

---

## [Author Response · Author response to Decision Letter 0]

27 Oct 2021

Response to Editors and Reviewers

I would like to thank the Editors and both Reviewers very much for their time devoted to reviewing the manuscript. Thank you also very much for the valuable comments in the reviews.

Journal Requirements:

1. The required names of the submitted files were introduced according to the recommendations of the Editors and Reviewers.

2. The Methods section provides additional details on the experimental materials (soil and sewage sludge) that were used for the study.

3. The source of funding for the research and publication comes from the Ministry's funds for scientific research of the Częstochowa University of Technology and it is necessary to provide the information in Acknowledgments: „The scientific research was funded by the statute subvention of Czestochowa University of Technology, Faculty of Infrastructure and Environment”. Such an entry in the publication is necessary for us to receive a refund of the cost of the publication. It is required at the Częstochowa University of Technology. What should I do then? As recommended, I removed Acknowledgments. I am asking You to make the necessary changes on my behalf.

5. Answer as in points 3 and 4

6. The manuscript includes the results of the research along with the dataset underlying the findings presented. The data on drug resistance in the tables are specific and therefore presented in this extended form. Therefore, a separate link to the data is not included. They are available in the manuscript. Based on the data contained in the publication, it is possible to reproduce and verify the entire research process.

7. I logged into Editorial Manager with my ORCID ID. I filled in the details.

Responses to Reviewers

Reviewer 1

Thank you for your time and valuable comments in your review.

Changes to the text are highlighted in yellow.

Recommended revisions have been made to the Abstract, Introduction to the study, and Research Methodology.

Answer regarding Tables 3 and 4: in the tables, the amounts of Escherichia coli, Clostridium perfringens and Proteus vulgaris bacteria are given in the form of so-called bacterial titer. The table incorrectly used the description "Bacteria count". The incorrect description has been corrected. 

The microbial titer is the smallest volume of test material in which there is at least one living cell of the indicator microorganism. The titer determination is used to determine the degree of microbial contamination of the test material. The coliform titer test is most commonly performed because it is a bacterium that is found in human and animal feces. The coliformtiter determination is the primary method for assessing whether water or another environment (soil) is contaminated with feces.The coliform titer is different from another indicator called the coliform index. The coliform index provides the number of coliform bacteria in 1 dm³. In determining the index, cultures made on Endo medium agar and colonies identified as coliform are counted.

The number of colony-forming bacteria units in the tables described as "Total bacteria count [CFU/1ml]" was determined for the total count of mesophilic bacteria,the Enterococcus and Enterobacteriaceae family.

An increased dose of sewage sludge applied to the soil fertilization caused an increase in the contamination with Escherichia coli of the fertilised soil and the water infiltrating it.

Answer concerning Tables 5, 6, 7 and 8: the results of drug resistance tests are related to various pathogenic groups of microorganisms from different environments tested. Tables 6 - 8 also show which groups of potentially pathogenic microorganisms were found after application of which doses of fertilizing sewage sludge. Data from Tables 5 -7 are compared in the description of the study results. The Reviewer's valuable comments on the form of presentation will be used in planning the next experiments. 

Reviewer 2

Thank you very much for your valuable comments on our manuscript. I would also like to thank the Reviewer for appreciating the really large amount of organizational and research work.

Changes suggested and recommended by the Reviewer have been made to the paper.

Changes to the text are highlighted in green.

Abstract line 13-23

A brief characterization of the main results has been added in the Abstract.

Materials and Methods line 99-100

The methodology for sanitary analysis determination has been complemented

Materials and Methods line 135

Corrected as recommended

Results Table 1. line 168-171

The horizontal lines in Table 1 have been added.

Results line 223-225

This part has been complemented

Results line 231-233

The sentence has been removed

Results Table 6. and Table 7. line 240 and 252

The values presented in these tables represent sewage sludge application rates after which resistant bacteria occurred in the aquatic and soil environments.As suggested by the Reviewer, an additional explanatory description has been added below the table.

Discussion line 317-320

Thank you for finding the mistake. A previous value was incorrectly repeated. A correction has been made to the text.

Discussion line 322

A correction has been made to the text.

Manuscript

Units has been checked.

---

## [Decision Letter · Decision Letter 1]

22 Nov 2021

PONE-D-21-26629R1Analysis of migration of pathogenic drug-resistant bacteria to soils and groundwater after fertilization with sewage sludgePLOS ONE

Dear Dr. Stańczyk-Mazanek,

Thank you for submitting your manuscript to PLOS ONE. After careful consideration, we feel that it has merit but does not fully meet PLOS ONE’s publication criteria as it currently stands. Therefore, we invite you to submit a revised version of the manuscript that addresses the points raised during the review process.Please do address the minor comment of Reviewer 2 as it is required for acceptance.Please submit your revised manuscript by Jan 06 2022 11:59PM. If you will need more time than this to complete your revisions, please reply to this message or contact the journal office at plosone@plos.org. Please include the following items when submitting your revised manuscript:A rebuttal letter that responds to each point raised by the academic editor and reviewer(s). You should upload this letter as a separate file labeled 'Response to Reviewers'.A marked-up copy of your manuscript that highlights changes made to the original version. You should upload this as a separate file labeled 'Revised Manuscript with Track Changes'.An unmarked version of your revised paper without tracked changes. You should upload this as a separate file labeled 'Manuscript'.If applicable, we recommend that you deposit your laboratory protocols in protocols.io to enhance the reproducibility of your results. Protocols.io assigns your protocol its own identifier (DOI) so that it can be cited independently in the future. For instructions see: https://journals.plos.org/plosone/s/submission-guidelines#loc-laboratory-protocols. Additionally, PLOS ONE offers an option for publishing peer-reviewed Lab Protocol articles, which describe protocols hosted on protocols.io. Read more information on sharing protocols at https://plos.org/protocols?utm_medium=editorial-email&utm_source=authorletters&utm_campaign=protocols.

We look forward to receiving your revised manuscript.

Kind regards,

Reginald B. Kogbara, Ph.D.

Academic Editor

PLOS ONE

Journal Requirements:

Reviewers' comments:

Reviewer's Responses to Questions

**Comments to the Author**

1. If the authors have adequately addressed your comments raised in a previous round of review and you feel that this manuscript is now acceptable for publication, you may indicate that here to bypass the “Comments to the Author” section, enter your conflict of interest statement in the “Confidential to Editor” section, and submit your "Accept" recommendation.

Reviewer #1: All comments have been addressed

Reviewer #2: All comments have been addressed

2. Is the manuscript technically sound, and do the data support the conclusions?

Reviewer #1: Yes

Reviewer #2: Yes

3. Has the statistical analysis been performed appropriately and rigorously? 

Reviewer #1: Yes

Reviewer #2: Yes

4. Have the authors made all data underlying the findings in their manuscript fully available?

Reviewer #1: Yes

Reviewer #2: Yes

5. Is the manuscript presented in an intelligible fashion and written in standard English?

Reviewer #1: Yes

Reviewer #2: (No Response)

6. Review Comments to the Author

Reviewer #1: Author could have done better in addressing review question but I believe the author efforts are sufficient enough.

Reviewer #2: I am grateful for answering every comment and making corrections. Unfortunately, one of the comments has been wrongly corrected. After solving this issue, the article in my opinion is ready for publication. Therefore, the issue is described below.

- In the revised manuscript Authors removed the sentence in line 260; “A worrying phenomenon was the observed resistance of E. faecalis bacteria to erythromycin.”.

In the Review, I had in mind the other sentence, which is: “It was found that only a dose of 10 t/ha of sewage sludge did not cause significant contamination of sandy soil and infiltrating water.”. I allow myself to recall my comment for this sentence: “The last sentence of the paragraph is unclear and it could be clarified with references to specific Table/Tables.”.

7. PLOS authors have the option to publish the peer review history of their article (what does this mean?). If published, this will include your full peer review and any attached files.

Reviewer #1: No

Reviewer #2: No

---

## [Author Response · Author response to Decision Letter 1]

23 Nov 2021

Dear Reviewer,

Thank You very much for Your valuable comments on our work.

A correction has been made to the text.

Changes to the text are highlighted in green.

Yours faithfully, 

Ewa Stańczyk-Mazanek

---

## [Editor Report · Decision Letter 2]

25 Nov 2021

Analysis of migration of pathogenic drug-resistant bacteria to soils and groundwater after fertilization with sewage sludge

PONE-D-21-26629R2

Dear Dr. Stańczyk-Mazanek,

We’re pleased to inform you that your manuscript has been judged scientifically suitable for publication and will be formally accepted for publication once it meets all outstanding technical requirements.

Kind regards,

Reginald B. Kogbara, Ph.D.

Academic Editor

PLOS ONE
---

## [Editor Report · Acceptance letter]

6 Dec 2021

PONE-D-21-26629R2 

Analysis of migration of pathogenic drug-resistant bacteria to soils and groundwater after fertilization with sewage sludge 

Dear Dr. Stańczyk-Mazanek:

I'm pleased to inform you that your manuscript has been deemed suitable for publication in PLOS ONE. Congratulations! Your manuscript is now with our production department. 

Kind regards, 

on behalf of

Dr. Reginald B. Kogbara 

Academic Editor

PLOS ONE